# The Atypical Cyclin-Dependent Kinase 5 (Cdk5) Guards Podocytes from Apoptosis in Glomerular Disease While Being Dispensable for Podocyte Development

**DOI:** 10.3390/cells10092464

**Published:** 2021-09-18

**Authors:** Nicole Mangold, Jeffrey Pippin, David Unnersjoe-Jess, Sybille Koehler, Stuart Shankland, Sebastian Brähler, Bernhard Schermer, Thomas Benzing, Paul T. Brinkkoetter, Henning Hagmann

**Affiliations:** 1Center for Molecular Medicine Cologne, Department II of Internal Medicine, Faculty of Medicine-University Hospital Cologne, University of Cologne, 50937 Cologne, Germany; nicole.mangold1@uk-koeln.de (N.M.); david.unnersjoe-jess@uk-koeln.de (D.U.-J.); v1skoehl@ed.ac.uk (S.K.); sebastian.braehler@uk-koeln.de (S.B.); bernhard.schermer@uk-koeln.de (B.S.); thomas.benzing@uk-koeln.de (T.B.); 2Division of Nephrology, University of Washington, Seattle, WA 98195, USA; scoobie@uw.edu (J.P.); stuartjs@uw.edu (S.S.); 3Cologne Cluster of Excellence on Cellular Stress Responses in Ageing-Associated Diseases (CECAD) and Systems Biology of Ageing Cologne (Sybacol), University of Cologne, 50935 Cologne, Germany

**Keywords:** proteinuria, nephrotoxic nephritis, apoptosis, cyclin I, p35

## Abstract

Cyclin-dependent kinase 5 (Cdk5) is expressed in terminally differentiated cells, where it drives development, morphogenesis, and survival. Temporal and spatial kinase activity is regulated by specific activators of Cdk5, dependent on the cell type and environmental factors. In the kidney, Cdk5 is exclusively expressed in terminally differentiated glomerular epithelial cells called podocytes. In glomerular disease, signaling mechanisms via Cdk5 have been addressed by single or combined conventional knockout of known specific activators of Cdk5. A protective, anti-apoptotic role has been ascribed to Cdk5 but not a developmental phenotype, as in terminally differentiated neurons. The effector kinase itself has never been addressed in animal models of glomerular disease. In the present study, conditional and inducible knockout models of Cdk5 were analyzed to investigate the role of Cdk5 in podocyte development and glomerular disease. While mice with podocyte-specific knockout of Cdk5 had no developmental defects and regular lifespan, loss of Cdk5 in podocytes increased susceptibility to glomerular damage in the nephrotoxic nephritis model. Glomerular damage was associated with reduced anti-apoptotic signals in Cdk5-deficient mice. In summary, Cdk5 acts primarily as master regulator of podocyte survival during glomerular disease and—in contrast to neurons—does not impact on glomerular development or maintenance.

## 1. Introduction

The atypical cyclin-dependent kinase 5 (Cdk5) is a proline-directed serine/threonine kinase with prime functions in terminally differentiated cell types [1]. In neurons, Cdk5 is a crucial regulator of brain development and morphological characteristics, including neuronal migration, axonal guidance, and synapse formation and plasticity, as well as cytoskeletal rearrangement. Cdk5 is involved in higher cognitive abilities, e.g., learning, memory and behavior [2,3,4,5]. The kinase is activated by specific activators, such as p35, p25, p39, and cyclin I, and inhibited by GSTP1, cyclin D1, and cyclin E [6,7,8,9,10,11]. These activators and inhibitors confer a tight spatial and temporal regulation of Cdk5 activity [12,13,14]. In mice, the conventional knockout of Cdk5 is associated with severe structural lesions of the central nervous system and perinatal mortality [15].

In the kidney, Cdk5 is exclusively expressed in glomerular epithelial cells called podocytes [16]. Podocytes are postmitotic epithelial cells arrested in G0/1 phase that reside on the outer aspect of glomerular capillaries. Podocytes protrude primary and secondary cellular processes, called foot processes, which interdigitate with foot processes of neighboring cells. Podocyte foot processes compress the glomerular basement membrane against positive capillary pressure and regulate mesh width of the glomerular basement membrane to limit the passage of macromolecules, specifically proteins larger than 70 kDa [17]. Loss of podocytes either by detachment from the glomerular basement membrane or by apoptosis cannot be replaced by proliferation, which results in loss of function of the glomerular filter, indicated clinically by urinary protein loss (i.e., proteinuria/albuminuria) and histologically by glomerular scarring.

In podocytes, Cdk5 is activated by p25, p35, and/or cyclin I, which also determine the subcellular localization of the kinase [18]. Depending on the binding partners, Cdk5 exerts an anti-apoptotic signal via BCL-2/BCL-XL through diverse mechanisms [7,19]. Loss of Cdk5 activation, by conventional knockout of p35 or cyclin I or both, in mice was associated with higher susceptibility to injury in the mouse model of nephrotoxic nephritis (NTN) [20,21]. In these studies, cyclin I-Cdk5 was shown to activate ERK 1/2, leading to increased BCL-2/BCL-XL mRNA expression, whereas p35-Cdk5 phosphorylated Bcl-2 directly, resulting in stabilization of the protein complex. In addition, p35-Cdk5 phosphorylates the non-selective cation channel TRPC6, thereby increasing channel activity [22]. Enhanced conductance across TRPC6 is associated with podocyte dysfunction in genetic and sporadic diseases [23,24,25]. As both inhibition and promotion of Cdk5 activity may be detrimental to the podocyte, a tight control of Cdk5 kinase activity is warranted in these highly specialized glomerular epithelial cells. Strikingly, knockout of the Cdk5-activators, p35 or cyclin I or both, did not affect podocyte development or function under non-stressed conditions. In contrast, following podocyte stress in the nephrotoxic nephritis model (NTN), apoptosis was higher in the p35/cyclin I double knockout mice compared to a similar stress in the single p35 or cyclin I null mice.

To investigate a potential developmental role of Cdk5, which may be referred by yet unknown activators of the kinase, we generated podocyte-specific Cdk5 knockout mice and analyzed these animals under non-stressed conditions. In addition, we delineated the contribution of Cdk5 in states of glomerular disease, employing inducible podocyte-specific Cdk5 knockout mice and the NTN disease model.

## 2. Materials and Methods

### 2.1. Animal Models

Podocyte-specific Cdk5 knockout was generated by mating Cdk5flox mice obtained from Jackson Laboratory (Bar Harbor, ME, USA) with hNPHS2.Cre mice [1,26]. Transgenic offspring were identified by PCR amplification using specific primer sequences listed below (Table 1). 

For the assessment of Cdk5 knockout efficacy, Cdk5flox mice were mated with mPodocin.2A.Cre.2A.mTomato mice, yielding mTomato-fluorescent podocytes in the offspring [3]. Glomeruli were isolated and primary podocytes purified by fluorescence-activated cell sorting (FACS), as previously described [27,28,29]. Cdk5 expression levels were quantified in isolated podocytes by quantitative PCR (primer sequences in Table 1). 

In all animal studies, only male mice were employed. Mice were bred into mixed FVB/N and 129S4/SvJae backgrounds and maintained under standardized, pathogen-free conditions in the University of Washington animal facility, as well as in the University of Cologne animal facility. The experimental protocol was reviewed and approved by the Animal Care Committee of the University of Washington, Seattle, USA and by the State Agency for Nature, Environment, and Consumer Protection, North Rhine-Westphalia, Germany (Landesamt für Natur, Umwelt und Verbraucherschutz Nordrhein-Westfalen).

### 2.2. Generation of Inducible Cdk5 Knockout

Temporal control of Cdk5 knockout in podocytes was achieved by mating Cdk5flox mice with hNPHS/rtTA/TetO-Cre mice. In the offspring, transgene transcription was controlled by a podocin-driven promoter which expressed Cre recombinase specifically in podocytes in the presence of doxycycline [30]. Doxycycline (Sigma/Merck, Darmstadt, Germany) was administered for 14 days via the drinking water (0.2 mg/mL in 5% sucrose), between the ages of 8 and 10 weeks. Water was exchanged twice weekly, and the bottles were protected from light to prevent doxycycline degradation. Transgenic mice were identified by PCR amplification. Primer sequences are listed in the following table.

### 2.3. Experimental Glomerular Disease 

Nephrotoxic nephritis was induced by intraperitoneal injections of sheep anti-rabbit glomerular antibody, as previously described [31]. Fourteen days after the induction of Cdk5-deletion in podocytes, nephrotoxic serum (20 mg/20 g body weight) was injected intraperitoneally into 10–12 week-old mice on two consecutive days. Mice were sacrificed on day 7 after the second injection.

### 2.4. Immunohistochemistry

Immunohistochemical analyses were performed on kidney sections of podocyte-specific Cdk5-knockout (Cdk5^pko^) and wild type mice and stained with primary antibodies listed in the Table 2. Briefly, formaldehyde-fixed, paraffin-embedded kidneys were cut into 4 µm tissue sections, deparaffinized in xylene, and rehydrated in graded alcohol. Antigen retrieval was performed by boiling kidney sections for 10 min in 10 mmol/L citrate buffer, pH 9. Endogenous peroxidase activity was blocked with 3% hydrogen peroxide, and endogenous biotin was inhibited by Avidin/Biotin blocking kit (Vector Laboratories, Burlingam, CA, USA). Kidney sections were incubated overnight at 4 °C, with the respective primary antibody diluted in 1% PBS/BSA buffer. Subsequently, biotin-conjugated anti-rabbit secondary antibody (Jackson ImmunoResearch) was diluted in 1% PBS buffer and incubated for one hour at room temperature. Repeated washing steps were performed with PBS buffer at least three times. The ABC kit (Vector Laboratories, Burlingam, CA, USA) was used for signal amplification. Chromogen 3,3′-diaminobenzamine (DAB; Sigma, St. Louis, MO, USA) was used. Finally, sections were counterstained with hematoxylin (Sigma Aldrich, St. Louis, MO, USA), dehydrated in xylol, and mounted with Histomount (National Diagnostics, Atlanta, GA, USA). 

### 2.5. Quantitative Assessment of Podocyte Number and Apoptosis

Podocyte number was quantified on stained paraffin-embedded kidney sections with a specific primary antibody against Wilms’ tumor 1 protein (WT1) [4]. Six animals of each genotype were analyzed in a blinded fashion by counting WT1-positive cells in, at a minimum, 50 glomeruli per section. Apoptotic cells were quantified by immunohistochemical staining for cleaved caspase-3 on formaldehyde-fixed kidney sections. Cleaved caspase-3-positive cells per glomerulus were counted in at least 50 glomeruli per animal of each cohort (inducible podocyte-specific Cdk5-knockout (Cdk5^ipko^), wt; *n* = 6 each). 

### 2.6. Evaluation of Glomerulosclerosis

Glomerulosclerosis was quantified on paraffin-embedded kidney sections stained with periodic acid-Schiff reagent as percentage of involved glomeruli per representative section, in all individual groups (for each, *n* = 6). In addition, the glomerulosclerosis index was assessed. To this end, 50 glomeruli minimum per individual animal were categorized based on their percentage of scarred glomerular tuft area, divided into 4 groups: grade 1, <25%; grade 2, 25–50%; grade 3, 50–75%; and grade 4, 75–100% [7,22]. 

### 2.7. Imaging

All stained tissue sections were scanned with a slide scanner for brightfield images Leica SCN400 (Leica Biosystems, Wetzlar, Germany) and further assessed with ImageScope (Aperio) image-processing software v12.1 (Leica Biosystems, Wetzlar, Germany).

### 2.8. STED Imaging and Computed Analyses

A simplified version of a previously published protocol was used [32]. Formaldehyde-fixed kidneys were embedded in 3% agarose in DI water and sectioned to 200 µm thickness using a vibratome. Slices were then incubated in clearing solution (200 mM boric acid, 4% SDS, pH 8.5) at 50 °C overnight. Sections were washed in PBST (0.1% Triton-X in 1X PBS, pH 7.4) for 10 min before incubation in a sheep polyclonal antibody against nephrin (R&D systems, Minneapolis, MI, USA, AF4269) diluted 1:50 in 10 mM HEPES, pH 7.5 with 200 mM NaCl and 10% TritonX-100 at 37 °C for 4 h with shaking at 500 rpm. After primary antibody incubation, samples were washed in PBST for 5 min at 37 °C and were then incubated in a donkey anti-sheep secondary antibody conjugated to Abberior STAR635P (Abberior, Goettingen, Germany, 2-0142-007-2, dilution 1:50) at 37 °C for 4 h. Samples were incubated in 80% *wt*/*wt* fructose (1 mL of dH_2_O added to 4 g of fructose) at 37 °C for 15 min and then mounted in a MatTek dish with a cover slip on top, prior to imaging with a Leica SP8 3× gSTED system (Leica Biosystems, Wetzlar, Germany) using a 100× 1.4 NA objective. To quantify the slit diaphragm length per area, a previously published ImageJ macro was used [17].

### 2.9. Measurement of Proteinuria 

For the assessment of proteinuria, spot urine was collected from animals of each cohort at day 0, as baseline, and at the following days as indicated. Initial analyses were performed by Coomassie blue staining after SDS-PAGE of small volumes of urine. Quantitative assessment was performed by measuring urinary protein using the sulfosalicylic acid method [33] and urinary creatinine using the Creatinine Colorimetric Assay Kit (Cayman Chemical, Ann Arbor, MI, USA). Assays were performed according to manufacturers’ instructions. 

### 2.10. Statistics

Statistical calculations were performed with PalmPrism. For histological assessment, Image Scope (Aperio Version 12.1, Leica Biosystems, Wetzlar, Germany) was used.

## 3. Results

### 3.1. Podocyte-Specific Knockout of Cdk5 Shows Regular Glomerular Histology and Function

Conditional podocyte-specific Cdk5 knockout mice (Cdk5^pko^) were generated by crossing mice carrying the floxed Cdk5 allele with mice containing Cre recombinase under the podocyte-specific human podocin promoter (Figure 1A). In Cdk5flox mice, the first five exons are flanked by loxP sites. Cdk5 expression was determined by qPCR on samples of primary podocytes isolated from mice carrying heterozygous hNPHS.Cre and homozygous Cdk5-floxed (Cdk5^pko^), heterozygous Cdk5-floxed (Cdk5^het^), or homozygous wild type Cdk5 (Cdk5^wt^) alleles. Expression of Cdk5 was reduced to 46.31% (SD 11.86) in Cdk5^het^ and 13.1% (SD 2.26) in Cdk5^pko^ mice, compared to Cdk5 wild type littermates (Figure 1B).

Cdk5^pko^ mice were born in a regular Mendelian ratio with no apparent developmental defects. Life expectancy was not reduced compared to Cdk5 wild type littermates (data not shown). Phenotyping included screening for proteinuria, and kidney histology was performed in Cdk5^pko^ mice at 70 weeks of age. Coomassie staining of urine samples showed no detectable proteinuria, specifically, no albuminuria (Figure 2A). PAS- and AFOG-staining revealed normal kidney morphology without evidence of glomerular disease (Figure 2B). The regular, garland-like immunostaining pattern of podocin, tracing the glomerular slit diaphragm, was similarly detected in Cdk5^pko^ and Cdk5^het^ mice (Figure 2C). Podocyte number per glomerulus, assessed by staining for WT1, was similar in Cdk5^pko^ and heterozygous control mice (Figure 2C,D). To exclude subtle structural defects of foot process architecture at the nanoscale, we utilized STED-imaging on cleared kidney tissue after immunofluorescent labeling of nephrin (Figure 2E). STED-images revealed an evenly distributed podocin signal and a regular pattern of foot processes in all samples. Computed analysis of STED-images did not identify significant differences in slit diaphragm length between Cdk5^pko^, Cdk5^het^, and Cdk5^wt^ littermates (Figure 2F).

### 3.2. Podocyte-Specific Knockout of Cdk5 Shows Higher Susceptibility to Glomerulosclerosis and Reduced Kidney Function in the Nephrotoxic Nephritis Model

We next assessed the biological impact of podocyte Cdk5 deficiency in glomerular disease. To exclude undetected developmental aberrations following Cdk5 deficiency, or compensatory mechanisms that might be activated in the absence of Cdk5 during development, inducible Cdk5 knockout mice (Cdk5^ipko^) were generated (Figure 3A). In these mice, activity of Cre recombinase leading to knockout of Cdk5 was induced at the age of 8–10 weeks by administration of doxycycline in the drinking water over 14 days (Figure 3B). Baseline urine samples were obtained before doxycycline administration and bi-weekly thereafter up to day 56. Quantitative analysis of urinary protein and creatinine revealed no relevant proteinuria at baseline (Figure 3C).

Experimental glomerular disease was induced after doxycycline-dependent Cdk5 knockout by intraperitoneal injection of nephrotoxic serum (NTS) in mice 10 weeks of age (Figure 4A). Histological analysis and quantification of proteinuria were performed on day 7 after induction of glomerular disease. PAS- and AFOG-staining revealed glomerular sclerosis in Cdk5^ipko^ as well as control animals (Figure 4B). Glomerular scarring was greater in Cdk5 knockout compared to control animals (mean grade Cdk5^ipko^: 1.79 (SD 0.10) vs. Cdk5^contr^: 1.37 (SD 0.16); *p* < 0.05) (Figure 4C). In addition, the extent of glomerular involvement (percentage of glomeruli with sclerosis) was quantified at 56% (SD 5) in Cdk5 knockout mice, as compared to 30% (SD 7) in control mice (Figure 4D). No glomerular damage was detected in control Cdk5^ipko^ and Cdk5^contr^ given vehicle (saline) (Figure 4C,D). Nephrotic range proteinuria detected in both mouse strains at day 7 of disease was higher in Cdk5^ipko^ mice (Cdk5^ipko^: 62.46 *g*/*g* (SD 15.35) vs. Cdk5^contr^: 22.11 *g*/*g* (SD 5.10); *p* < 0.05). No significant proteinuria was detected in control saline-treated mice (Figure 4E).

### 3.3. Cdk5-Deficient Mice Show Higher Rate of Apoptosis and Decreased Anti-Apoptotic Signal

Previously, we and others established the pro-survival signals referred by Cdk5 in podocytes and other post-mitotic cells [7,19,20,34]. In podocytes, cyclin I-Cdk5 activates MAP kinases MEK1/2, followed by promotion by ERK1/2 of the expression of the anti-apoptotic genes BCL-2 and BCL-XL. In addition, p35-Cdk5 phosphorylates the Bcl-2 protein to enhance its stability [7].

Podocyte apoptosis, quantitated by cleaved caspase-3 (CC3) staining, was not detected in Cdk5^ipko^ mice or controls prior to disease induction (day 0) (Figure 5A,B). At day 7 of disease, podocyte apoptosis was threefold higher in Cdk5^ipko^ mice compared to control mice. At baseline, there was no significant difference in Bcl-2 staining in Cdk5^ipko^ mice compared to control, although, in Cdk5 knockout mice, a trend to less positive cells was noted (Figure 5C,D). In diseased Cdk5^ipko^ mice, Bcl-2 staining was significantly lower compared to Cdk5 control mice. For Bcl-XL, however, already at baseline, a significant reduction of staining-positive glomerular cells was detected in Cdk5 knockout as compared to control (Figure 5E,F). Even though they increased in both control and Cdk5^ipko^ mice after NTS challenge, Bcl-XL positive cells were more abundant in control compared to Cdk5 knockout tissue.

## 4. Discussion

Cdk5 is an ubiquitously expressed proline-directed serine/threonine kinase implicated in both physiological and pathological cellular processes. Amongst these are cytokine production, regulation of insulin levels, migration and invasion, angiogenesis, myogenesis, apoptosis, and senescence (reviewed in Sharma and Sicinski [35]). Importantly, tight control of Cdk5 activity is crucial since both hyperactivity and inhibition of Cdk5 result in cellular dysfunction. Noteworthy is that cell-specific regulation of Cdk5 activity is controlled by cell type-specific activators. For example, in post-mitotic neurons, p35 and p39 are the major regulators of Cdk5 activity. Combined loss of p35 and p39 mimics the phenotype of Cdk5 deficiency, with perinatal lethality due to defective development of the brain [15,36]. In contrast, mice with a combined deletion of both p35 and cyclin I develop normally and show no phenotype under non-stressed conditions [20]. The results of the present study show that podocyte-specific deletion of Cdk5 does not impact development. In addition, even in advanced age, Cdk5 deficiency in podocytes is not associated with a pathological phenotype under non-stressed conditions. These results are in striking contrast to the severe phenotype in neuronal cells following CDK5 deletion [37,38]. This finding comes not without surprise since the podocytes’ branching morphology, supported by a microtubular lattice and F-actin-based foot processes, shares high similarity with the cytoskeleton of dendrite-forming neurons [39,40]. By phosphorylation of Rho-GTPases, Cdk5 plays a crucial role in controlling actin cytoskeletal modification and synaptic plasticity in neuronal cells [41,42]. In podocytes, however, highly active regulation of the actin cytoskeleton appears to be independent of Cdk5 activity and is, therefore, not affected by Cdk5 deficiency, neither during glomerulogenesis nor later in life. Quantification of podocytes per glomerular cross section was performed on the basis of immunostaining for WT1. WT1 staining detects mature podocytes and is not the ideal marker in disease states when podocytes de-differentiate and lose WT1 reactivity. In this study employing non-stressed mice, however, WT1 staining reliably estimated podocyte number. Consistent podocyte number in Cdk5^pko^ vs. control, reflected by the regular glomerular architecture and function, i.e., absence of proteinuria, argues against a role of Cdk5 in podocyte development.

Recently, several studies in podocytes investigated the beneficial role of the intermediate filament protein nestin in states of stress and its regulation of Cdk5/p35 inhibition of apoptosis [43,44,45]. The concept of apoptosis as a relevant mechanism of cell death in podocytes was often challenged over the past years [46,47]. A common critique arguing against apoptosis in podocytes concerns a lack of evidence of classical morphological signs of apoptosis, such as chromatin condensation, DNA fragmentation, and membrane blebbing in human glomerular disease or animal models. However, this argument disregards the three phases of apoptosis and the implication of the unique anatomical localization of podocytes in this context. The process of apoptosis progresses in three phases: induction, execution, and clearance. Caspase-mediated proteolysis, chromatin condensation, and DNA fragmentation occur during the execution phase. During the entire process of execution, plasma membrane integrity must be maintained to avoid the release of toxic waste products and injury to neighboring cells [48]. However, the first step in the execution phase is the release of focal adhesions and extracellular matrix interactions while actin rearranges to membrane-associated cortical rings [49,50,51]. For the podocyte, residing on the outer aspect of the glomerular tuft, release from the extracellular matrix and focal adhesions during the execution phase of apoptosis will result in detachment from the glomerular basement membrane and, also, from neighboring podocytes, followed by loss of podocytes in the urine. Therefore, podocytes showing classical apoptotic figures that appear later in the process of apoptosis will hardly be detected. In podocytes, the “point of no return”, which is usually mediated by permeability transition pores of the mitochondrial membrane and cytochrome c release, is reached during states of cytoskeletal rearrangement and lost ECM anchoring. This emphasizes the necessity of tight control of apoptosis in podocytes during the induction phase. The lack of any direct evidence for regulated, mitochondria-triggered forms of cell death in podocytes also raises the question of which intracellular signaling events control these pathways. Staining for cleaved caspase-3 and TUNEL staining are most commonly used to assess for apoptotic cells in tissue sections. However, TUNEL staining does not discriminate between apoptosis and other modes of cell death reliably [52]. Therefore, in the present study, apoptotic cells were detected by staining for cleaved caspase-3. We acknowledge that cleaved caspase-3 staining may underestimate the number of apoptotic podocytes, due to the loss of podocytes early in the process of apoptosis. Quantification of Bcl-2 and Bcl-XL expression was performed to assess the role of Cdk5 during initiation of apoptosis. Increased expression levels of both anti-apoptotic proteins recapitulated the mechanism proposed previously in which Cdk5/p35 increases BCL-2/BCL-XL expression and, in addition, stabilizes Bcl-2 at the protein level [7].

In conclusion, our study demonstrated that Cdk5 was not implicated in podocyte development but served a central role in the regulation of podocyte apoptosis. Cdk5 activity is crucial after toxic stimuli to avoid initiation of apoptosis, cytoskeletal remodeling, detachment from the glomerular basement membrane, and, eventually, loss of podocytes in the urine. Several studies on Cdk5 in neurological and oncological disorders showed good transferability of results gained in mice to human disease [53,54]. Consequently, it is tempting to speculate that activation of Cdk5 could be a novel therapeutic approach in human inflammatory glomerular disease.

## Figures and Tables

**Figure 1 cells-10-02464-f001:**
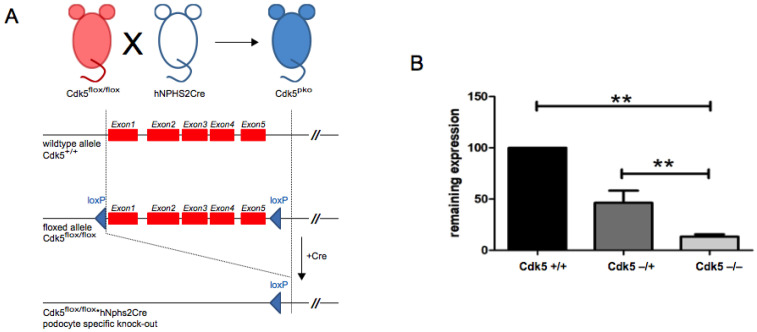
Breeding scheme and validation of podocyte-specific Cdk5 knockout. (**A**) Cdk5flox/flox mouse mated to the hNPHS2.Cre mouse yields podocyte-specific deletion of the Cdk5 gene (Cdk5^pko^). (**B**) Remaining expression of Cdk5 was quantified by qPCR from primary isolated podocytes. Cdk5 expression relative to wild type (Cdk5wt) is indicated. In podocyte-specific Cdk5flox-heterozygous mice (Cdk5^het^), Cdk5 expression was reduced to 46.31% (SD 11.86). For podocyte-specific knockout of Cdk5 (Cdk5^pko^), the remaining expression of Cdk5 was 13.1% (SD 2.26). ** *p*-value < 0.01, In each cohort, *n* = 3.

**Figure 2 cells-10-02464-f002:**
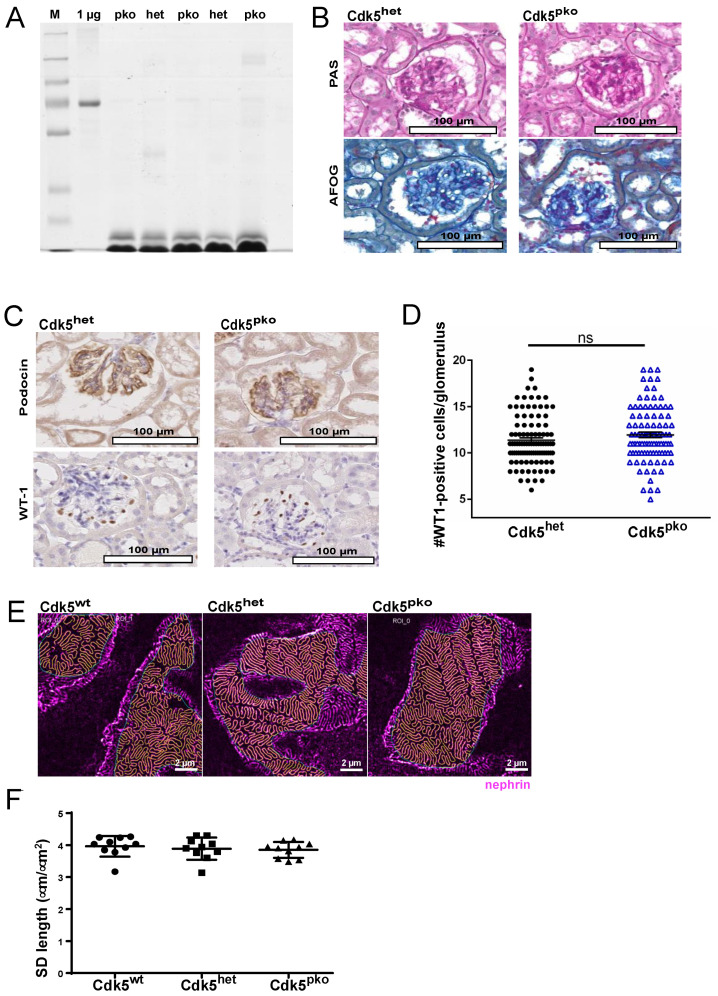
Baseline characterization of podocyte-specific Cdk5 knockout mice. (**A**): Coomassie gel of urine samples from Cdk5^het^ and Cdk5^pko^ mice at 70 weeks of age detects no relevant proteinuria. (**B**) PAS- and AFOG-staining of Cdk5^het^ and Cdk5^pko^ kidney sections show regular glomerular architecture. In each cohort, *n* = 6. (**C**) Expression of podocyte-specific markers podocin and WT1 is not affected by Cdk5 manipulation. (**D**) Podocyte number, quantified as WT1-positive cells per glomerulus, is not altered by podocyte-specific deletion of Cdk5. Dots: Cdk5^het^; triangles: Cdk5^pko^. In each cohort, *n* = 6. (**E**) Visualization of the slit diaphragm pattern on Cdk5^pko^, Cdk5^het^, and Cdk^wt^ kidney sections using STED microscopy on nephrin-stained kidney samples. Yellow lines denote the slit diaphragm segmented by the ImageJ macro. The blue line denotes the ROI within which the analysis is carried out (some minor areas with poor staining quality are excluded). Scale bar 2 µm. (**F**) SD length per area for the different genotypes. Each dot/square/triangle shows the SD length per area for one image (5 images per animal, *n* = 2 per genotype). Tukey’s multiple comparison test showed no significant difference between groups. Dots: Cdk5^wt^; squares: Cdk5^het^; triangles: Cdk5^pko^. Line represents mean and error bars represent standard deviation.

**Figure 3 cells-10-02464-f003:**
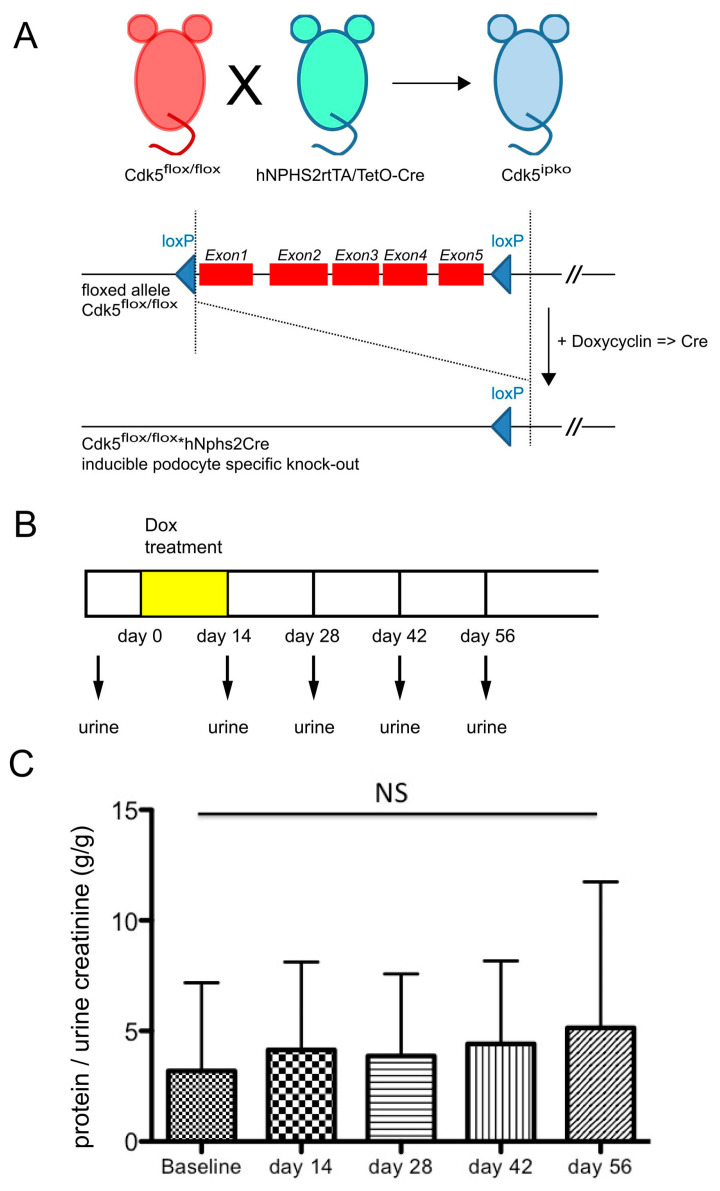
Mating scheme of inducible podocyte-specific knockout of Cdk5 and baseline characterization. (**A**) Cdk5flox/flox mouse mated to hNPHS2rtTA/TetO-Cre yields podocyte-specific doxycyline-inducible deletion of the Cdk5 gene (Cdk5^ipko^). (**B**) Schematic representation of doxycycline administration (14 days) and urine sampling in mice of 8–10 weeks of age. (**C**) Urine protein/creatinine ratio (*g*/*g*), followed over 56 days after initiation of doxycycline, is stable within normal limits. *n* = 6.

**Figure 4 cells-10-02464-f004:**
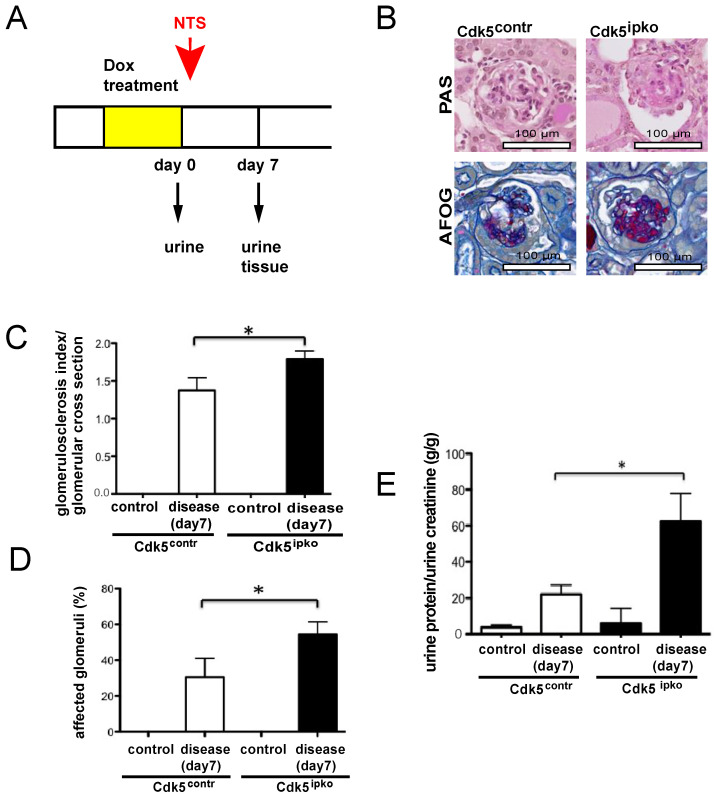
Nephrotoxic nephritis model in inducible podocyte-specific Cdk5 knockout mice. (**A**) Timing scheme of the NTN study including doxycycline treatment, nephrotoxic serum (NTS) application, urine sampling, and histologic analysis. (**B**) PAS- and AFOG-staining of Cdk5^contr^ and Cdk5^ipko^ kidney sections after NTS application reveals aggravated glomerular damage in Cdk5^ipko^. In each cohort, *n* = 6. (**C**) Quantification of glomerular sclerosis according to grades of percentage of the glomerular tuft area involved. The average glomerular fibrosis index per glomerular cross section was 1.79 (SD 0.10) for Cdk5^ipko^, as compared to 1.37 (SD 0.16) in Cdk5^contr^ after NTS application (* *p* < 0.05). No glomerular damage was detected in Cdk5^ipko^ and Cdk5^contr^ treated with saline instead of NTS. In each cohort, *n* = 6. (**D**) Extent of glomerular involvement quantified as percentage of glomeruli with sclerosis. Cdk5^ipko^ mice showed involvement of 56% (SD 5) of the glomeruli, whereas Cdk5^contr^ showed involvement of 30% (SD 7). In each cohort, *n* = 6; * *p* < 0.05. (**E**) Protein/creatinine ratio (*g*/*g*) of Cdk5^ipko^ compared to Cdk5^contr^ shows aggravated proteinuria of 62.46 (SD 15.35) *g*/*g* creatinine in Cdk5^ipko^ vs. 22.11 (SD 5.10) in Cdk5^contr^ (* *p* < 0.05). No significant proteinuria was detected in mice treated with saline instead of NTS. In each cohort, *n* = 6.

**Figure 5 cells-10-02464-f005:**
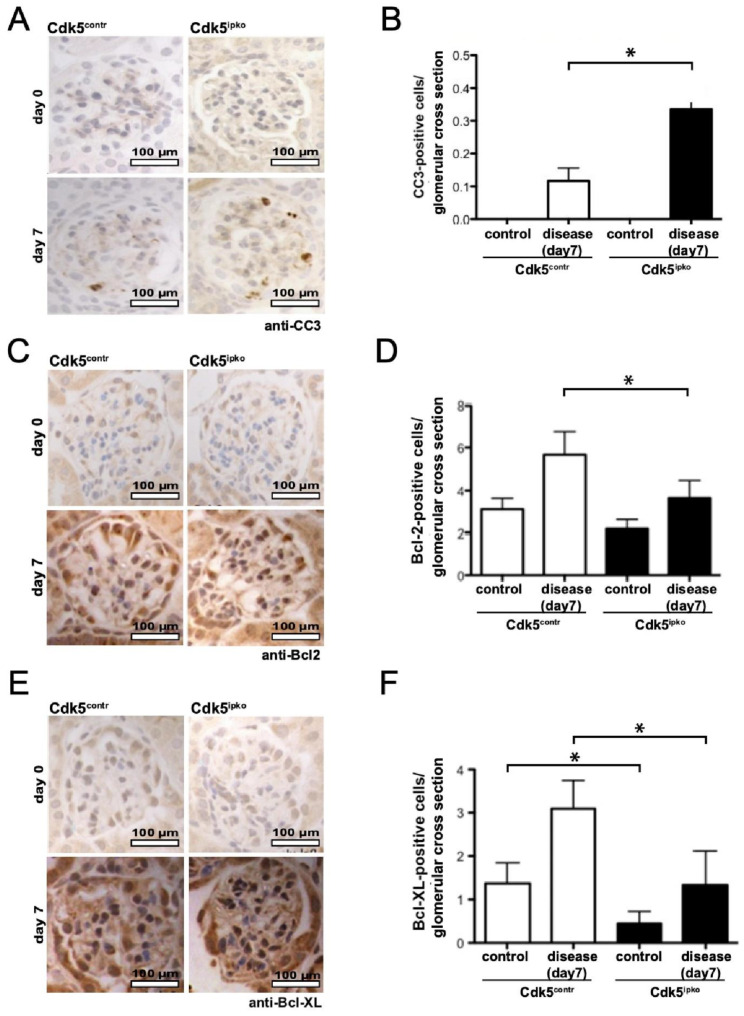
Analysis of pro- and anti-apoptotic signals dependent on podocyte expression of Cdk5 in the nephrotoxic nephritis model. (**A**,**B**): Detection of apoptotic cells by staining for cleaved caspase-3 (CC3) on kidney sections of Cdk5^contr^ and Cdk5^ipko^ mice. CC3-positive cells are more abundant in Cdk5^ipko^ as compared to Cdk5^contr^ (0.26 ± 0.05 vs. 0.12 ± 0.04; * *p* < 0.05). No CC3-positive cells were detected in mice treated with saline instead of NTS. In each cohort, *n* = 6. (**C**,**D**): Detection of anti-apoptotic Bcl-2 on kidney sections of Cdk5^contr^ and Cdk5^ipko^ mice. Increase in Bcl-2 expression after NTS challenge is lesser in Cdk5^ipko^ as compared to Cdk5^contr^ (3.7 ± 0.3 vs. 6.8 ± 0.5; * *p* < 0.05). Only few Bcl-2-positive cells were detected in mice treated with saline instead of NTS. In each cohort, *n* = 6. (**E**,**F**): Detection of anti-apoptotic Bcl-XL on kidney sections of Cdk5^contr^ and Cdk5^ipko^ mice. Increase of Bcl-XL expression after NTS challenge is lesser in Cdk5^ipko^ as compared to Cdk5^contr^ (1.3 ± 0.7 vs. 3.1 ± 0.5; *p* < 0.05). Expression of Bcl-XL was also reduced in Cdk5^ipko^ as compared to Cdk5^contr^ in the absence of NTN, i.e., after treatment with saline (0.4 ± 0.2 vs. 1.3 ± 0.3; * *p* < 0.05). In each cohort, *n* = 6.

**Table 1 cells-10-02464-t001:** Primer sequences for PCR amplification.

Mouse Model	Forward	Reverse
Cdk5^pko^	CAGTTTCTAGCACCCAACTGATGTA	GCTGTCCTGGAACTCCATCTATAGA
Cdk5mRNA (qPCR)	CAGTTTCTAGCACCCAACTGATGTA	GTCGTCCTGGAACTCCATCTATAGA
Cdk5^ipko^	GACCAGGTTCGTTCACTCA	TAGCGCCGTAAATCAAT

**Table 2 cells-10-02464-t002:** Primary antibodies for IHC analyses.

Name	Company	ID	Dilution
Bcl-2	Santa Cruz Biotechnology, Dallas, TX, USA	Sc-492	1:200
Bcl-XL	Cell Signaling Tech., Danvers, MA, USA	CST-2762	1:150
CC3	Cell Signaling Tech., Danvers, MA, USA	CST-9661	1:200
Podocin	Sigma, St. Louis, MO, USA	P0372	1:100
WT-1	Santa Cruz Biotechnology, Dallas, TX, USA	sc-393498	1:100

## Data Availability

Data are available upon request to corresponding authors.

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
