# Peer review of "The Atypical Cyclin-Dependent Kinase 5 (Cdk5) Guards Podocytes from Apoptosis in Glomerular Disease While Being Dispensable for Podocyte Development"

_cells, 2021, doi:10.3390/cells10092464_

Round 1
Reviewer 1 Report
The paper by Mangold et al. focuses on the cyclin-dependent kinase 5 (Cdk5) as a potential regulator of mouse glomerular podocyte phenotype and cytoskeleton, asking whether mutations/deletions might affect glomerular development and/or proteinuric glomerular disease. The answer is clearcut, since this knockout mouse model has no major abnormality of podocytes, contrary to neurons, who greatly resemble the branched architecture and ultrastructure of glomerular visceral epithelial cells. However, a role of Cdk5 may be found in the onset and progress of podocyte apoptosis upon independent, acquired injury to the kidney, such as in the classic nephrotoxic serum nephritis model. The study has a solid rationale and is technically powerful, although the results are far less exciting than expected, given the striking similarities of podocytes with neurons.
Major remarks:
- It is quite obvious that the Authors expected to detect kidney development abnormalities, proteinuria, glomerulosclerosis and eventually renal failure in these podocyte-specific Cdk5 KO mice. This did not happen, however. The only feature exhibited by Cdk5pko mice or their doxycycline-inducible counterpart was increased susceptibility to anti-GBM nephritis. This is somehow a “second-line” finding to save such labour-intensive project, which failed to identify a key regulator of podocyte phenotype and function.
- Along the same line, induction of another type of experimental proteinuric disease (e.g., Heymann or anti-Thy 1.1 nephritis) could strengthen the concept that Cdk5 knockout amplifies immune-mediated injury to the glomerulus, and hence inborn or acquired defects of the kinase may increase susceptibility to glomerular disease.
- Perhaps the issue should be considered from another point of view, that is, to see whether Cdk5 is downregulated or modified in other genetic or acquired proteinuric disease models (i.e. nephrin, podocin, TRPC6, WT-1 mutants or KO animals, 5/6 nephrectomy, etc.).
Author Response
Reviewer 1:
- It is quite obvious that the Authors expected to detect kidney development abnormalities, proteinuria, glomerulosclerosis and eventually renal failure in these podocyte-specific Cdk5 KO mice. This did not happen, however. The only feature exhibited by Cdk5pko mice or their doxycycline-inducible counterpart was increased susceptibility to anti-GBM nephritis. This is somehow a “second-line” finding to save such labour-intensive project, which failed to identify a key regulator of podocyte phenotype and function.
We thank the reviewer for this insightful comment. Indeed, we expected a developmental phenotype or aging phenotype when starting this project. We were surprised to find no abnormalities at baseline and therefore went on with a stress model to identify a role of Cdk5 in glomerular disease.
- Along the same line, induction of another type of experimental proteinuric disease (e.g., Heymann or anti-Thy 1.1 nephritis) could strengthen the concept that Cdk5 knockout amplifies immune-mediated injury to the glomerulus, and hence inborn or acquired defects of the kinase may increase susceptibility to glomerular disease.
We agree with the reviewer that other models of immune-mediated glomerular injury would underscore the role of Cdk5 in these diseases. However, due to the time limit for this revision, additional animal models are not realizable.
- Perhaps the issue should be considered from another point of view, that is, to see whether Cdk5 is downregulated or modified in other genetic or acquired proteinuric disease models (i.e. nephrin, podocin, TRPC6, WT-1 mutants or KO animals, 5/6 nephrectomy, etc.).
We reviewed Cdk5 regulation in previously published as well as unpublished proteomic datasets of animal models and podocyte cell culture. No regulation of Cdk5 in podocytes was detected in the lipopolysaccharide (LPS)-mouse model, in the puromycin aminonucleoside nephrosis (PAN) rat model, and in human samples of patients with Nephrin (n=2) and actinin 4 (n=2) mutations. A trend towards regulation of Cdk5 in podocytes was found in the adriamycin-induced nephropathy mouse model (-1.4-fold difference) and in PAN-treated human podocytes in culture (+ 0.3-fold difference). However, all results did not reach statistical significance (Rev.Fig.1; # = p-value > 0.05) [1,2].
RevFig.1
Due to the inconsistency and missing statistical significance of these data, we decided not to include them in the revised manuscript.
References:
- Koehler S, Kuczkowski A, Kuehne L, Jungst C, Hoehne M, Grahammer F, et al. Proteome Analysis of Isolated Podocytes Reveals Stress Responses in Glomerular Sclerosis. J Am Soc Nephrol. 2020;31(3):544-59.
- Rinschen MM, Grahammer F, Hoppe AK, Kohli P, Hagmann H, Kretz O, et al. YAP-mediated mechanotransduction determines the podocyte's response to damage. Sci Signal. 2017;10(474).

Reviewer 2 Report
Dear Authors,
The article submitted for review is fascinating. It should be emphasized that the effector kinase itself has never been studied in animal models of glomerular disease. In the study, the authors analyzed conditional and induced knockout models for Cdk5 to investigate the role of Cdk5 in podocyte development and glomerular disease. The authors showed that while the podocyte-specific Cdk5 knockout mice had no malformations and no regular lifespan, the loss of Cdk5 in podocytes increased the susceptibility to glomerular damage in a model of nephrotoxic nephritis. Glomerular damage was associated with reduced anti-apoptotic signals in Cdk5 deficient mice.
The researchers legitimately concluded that Cdk5 acts primarily as a major regulator of podocyte survival during glomerular disease and, unlike neurons, does not affect glomerular development or maintenance.
The study, based on which the manuscript was prepared, was properly planned and carried out. In my opinion, all studies are properly presented, and their results have been correctly interpreted.
Since it is a study in animal models, wouldn't it be better to modify the title a bit or actually add some clarification that these are studies on animal models, e.g. the title could be: The role of atypical Cyclin-dependent kinase 5 (Cdk5) in podocyte development and disease - the mice models study. I have a question whether the authors are sure that studies performed on human podocytes will show similar functions of Cdk5?
Besides, it would be interesting to refer to the practical aspect of the results obtained.
I believe that the article, after minor corrections, mainly linguistic, can be accepted for publication.
Best regards
Author Response
Reviewer 2:
- Since it is a study in animal models, wouldn't it be better to modify the title a bit or actually add some clarification that these are studies on animal models, e.g. the title could be: The role of atypical Cyclin-dependent kinase 5 (Cdk5) in podocyte development and disease - the mice models study.
We thank the reviewer for this valid point. We suggest changing the title of the manuscript to: “The atypical Cyclin-dependent kinase 5 (Cdk5) guards podocytes from apoptosis in glomerular disease in mice while being dispensable for podocyte development.”
- I have a question whether the authors are sure that studies performed on human podocytes will show similar functions of Cdk5?
Besides, it would be interesting to refer to the practical aspect of the results obtained.
We thank the reviewer for bringing up this important point. We have included a short paragraph on clinical implication of our findings at the end of the manuscript:
Several studies on Cdk5 in neurologic and oncologic disorders have shown good transfer-ability of results gained in mice to human disease [3,4]. Consequently, it is tempting to speculate, that activation of Cdk5 could be a novel therapeutic approach in human inflammatory glomerular disease.
Sincerely yours and for all authors,
Paul Brinkkötter and Henning Hagmann
References:
3. Allnutt AB, Waters AK, Kesari S, Yenugonda VM. Physiological and Pathological Roles of Cdk5: Potential Directions for Therapeutic Targeting in Neurodegenerative Disease. ACS Chem Neurosci. 2020;11(9):1218-30.
4. Sharma S, Zhang T, Michowski W, Rebecca VW, Xiao M, Ferretti R, et al. Targeting the cyclin-dependent kinase 5 in metastatic melanoma. Proc Natl Acad Sci U S A. 2020;117(14):8001-12.
